# A Comparison of the Efficacy of Antivenoms and Varespladib against the In Vitro Pre-Synaptic Neurotoxicity of Thai and Javanese Russell’s Viper (*Daboia* spp.) Venoms

**DOI:** 10.3390/toxins16030124

**Published:** 2024-03-01

**Authors:** Mimi Lay, Wayne C. Hodgson

**Affiliations:** Monash Venom Group, Department of Pharmacology, Biomedical Discovery Institute, Monash University, Clayton, VIC 3800, Australia; mimi.lay@monash.edu

**Keywords:** venom, snake, neurotoxicity, pre-synaptic, antivenom, Varespladib, Russell’s viper

## Abstract

The heterogeneity in venom composition and potency in disparate Eastern Russell’s viper (*Daboia siamensis*) populations has repercussions for the efficacy of antivenoms. This is particularly pronounced in geographical areas in which the venom of the local species has not been well studied and locally produced antivenoms are unavailable. In such cases, alternative therapies following envenoming, which are not limited by species specificity, may be employed to complement antivenoms. We studied the neuromuscular activity of *D. siamensis* venom from Thailand and Java (Indonesia) and the ability of Thai antivenoms and/or Varespladib to prevent or reverse these effects. Both Thai and Javanese *D. siamensis* venoms displayed potent pre-synaptic neurotoxicity but weak myotoxicity in the chick biventer cervicis nerve–muscle preparation. Whilst the neurotoxicity induced by both venoms was abolished by the prior administration of Thai *D. siamensis* monovalent antivenom or pre-incubation with Varespladib, Thai neuro-polyvalent antivenom only produced partial protection when added prior to venom. Pre-synaptic neurotoxicity was not reversed by the post-venom addition of either antivenom 30 or 60 min after either venom. Varespladib, when added 60 min after venom, prevented further inhibition of indirect twitches. However, the subsequent addition of additional concentrations of Varespladib did not result in further recovery from neurotoxicity. The combination of Thai monovalent antivenom and Varespladib, added 60 min after venom, resulted in additional recovery of twitches caused by either Thai or Javanese venoms compared with antivenom alone. In conclusion, we have shown that Varespladib can prevent and partially reverse the pre-synaptic neurotoxicity induced by either Thai or Javanese *D. siamensis* venoms. The efficacy of Thai *D. siamensis* monovalent antivenom in reversing pre-synaptic neurotoxicity was significantly enhanced by its co-administration with Varespladib. Further work is required to establish the efficacy of Varespladib as a primary or adjunct therapy in human envenoming.

## 1. Introduction

Snakes from the *Daboia* genus (Russell’s vipers) are widely, but discontinuously, distributed across South and South-East Asia. *Daboia russelii* is mainly found in Sri Lanka, Pakistan, Bangladesh, and southern India, whereas *Daboia siamensis* is mainly found in Myanmar; Taiwan; China; Thailand; Indonesia, including Eastern Java; and some of the lesser Sunda islands [1,2]. Most likely due to the large discontinuity across the Asiatic mainland and evolutionary processes, there is considerable variation at both the inter- and intra-species levels in the composition of venom, resulting in marked differences in symptoms in envenomed patients [3]. The potential impact of this on the effectiveness of antivenoms is noteworthy, primarily because uncommon or rare complications may emerge and could be inadequately addressed promptly. This is often attributed to a lack of awareness concerning specific conditions or the possibility of misdiagnosis [4]. In the context of Russell’s viper envenoming, the complexity of its multi-pathological effects poses a considerable challenge. Predicting the efficacy of antivenoms and discerning manifestations of symptoms becomes particularly problematic in such cases. This underscores the importance of heightened awareness and comprehensive understanding of the diverse and intricate nature of envenoming, ensuring a more effective response to potential complications.

Clinical symptoms of envenoming such as neurotoxicity, myotoxicity, cytotoxicity, haemotoxicity, coagulopathy, and renal damage vary due to differences in venom composition [5,6,7,8,9,10]. Clinically, whilst coagulopathy and nephrotoxicity are almost universal across the breadth of geographical distribution, there is significant variation in the neurotoxic effects caused by the venoms. For example, envenoming by *D. russelii* in Sri Lanka often results in additional neuromuscular paralysis and mild muscle damage, whereas neurotoxicity is largely absent in victims following *D. siamensis* envenoming in Myanmar [5,8,9], is tentatively suggested to be infrequent in Taiwan and China [11,12], and is seemingly very rare in Thailand [13]. However, the symptoms caused by *D. siamensis* in Indonesia are less well defined given there are limited case reports following bites from this sub-species.

The distribution of *D. siamensis* in Indonesia is disjunct, being absent from the Malayan Peninsula, Sumatra, Borneo, and most of Java, with less frequent sightings in Eastern Java, Komodo, Flores, and Lomblen [14]. The Indonesian species is quite isolated from other species, with almost 2,000 km separating them from the nearest conspecific populations in Thailand [15]. However, it has previously been shown that *D. siamensis* venoms from Thailand and Indonesia share similar venom profiles. Both display high levels of snake venom serine proteases (SVSP) (18.07% and 22.41%, respectively) and phospholipase A_2_ (PLA_2_) toxins (37.92% and 48.37%, respectively) despite their geographical separation [16]. Both Thai and Indonesian *D. siamensis* venoms are procoagulant in citrated human plasma [15] and can induce capillary leakage in mice [17]. However, the neuro-myotoxic effects of Indonesian and even Thai *D. siamensis* venom is less well defined than for some of the other geographical variants. For example, Thai *D. siamensis* venom has been shown to exert weak myotoxic effects in the chick biventer cervicis nerve–muscle preparation induced by both PLA_2_ and snake venom metalloproteases [18]. Although recent reports of envenoming from Indonesian species are not available, a previous publication on Russell’s viper systematics in Indonesia reported that neurotoxic effects were observed following envenoming in humans [1,14]. However, there was no pharmacological characterisation of the neurotoxic effects of Indonesian venom.

Thorpe et al. (2007) [9] suggested that the venom symptomology of *D. siamensis* in Thailand and Indonesia can only be tentatively predicted given that there is no clear connection between phylogenetic relationships and regional symptoms, especially in areas where the species has not been studied directly. This is certainly the case for Indonesian *D. siamensis* since the pharmacology of this venom has not been studied in detail. In contrast, the venoms of Sri Lankan *D. russelii* [8,17,19,20] and *D. siamensis* from China [21,22,23] and Thailand [15,16,18,23] have been more well studied.

Typically, neurotoxicity caused by *Daboia* venoms is due to the presence of PLA_2_ toxins, which represent a significant percentage of the total venoms [24,25,26]. Additionally, it has been reported that *D. siamensis* venom from Thailand contained similar neurotoxins to those found in Taiwanese *D. siamensis* venom, namely the pre-synaptic PLA_2_ toxins RV-4 and RV-7 [15,26]. However, despite the presence of neurotoxicity in in vivo animal models, neurotoxicity is not frequently reported following envenoming in humans by Taiwanese *D. siamensis* [12].

The absence of a specific antivenom raised against the venom of Indonesian *D. siamensis* necessitates the need for alternative treatments. The locally produced trivalent antivenom, SABU (Serum Anti Bisa Ular) lacks antibodies targeting *D. siamensis* venom toxins [27]. Consequently, SABU antivenom has been shown to be deficient for the cross-neutralisation of Indonesian *D. siamensis* venom [16,27]. Whilst *D. siamensis* monovalent antivenom from Thailand has been recommended for use against the venom of Indonesian *D. siamensis*, challenges with availability and access to this specific antivenom persist. Previously, Thai antivenoms raised against multiple species have been proven to be experimentally efficacious and clinically effective [15,16,17,28,29]. However, antivenoms need to be administered to snakebite patients as early as possible to increase their effectiveness [30]. There is also a need to search for alternative or adjunct therapies where the use of non-specific antivenoms is unavoidable or to complement antivenoms, particularly in instances where the administration of antivenom is delayed.

In recent years, small molecule inhibitors have been investigated for their use against toxins in snake venoms, particularly toxins that are ubiquitous across several snake species. LY315920, or Varespladib, is a specific inhibitor of PLA_2_ and has been previously shown to inhibit the PLA_2_ activity of 28 different species of snakes [31] and to increase survival, facilitate recovery, and delay the onset of symptoms in both elapid and viperid venom-induced toxicity [32,33,34,35]. Furthermore, Varespladib has been shown to prevent or reverse toxicity in in vitro preparations and in vivo mice and pig models of envenoming by venoms whose neurotoxic effects are predominantly due to pre-synaptic PLA_2_ neurotoxins, i.e., *Oxyuranus scutellatus* (Coastal taipan) and *Micrurus fulvius* (Eastern coral snake) [36,37,38]. Given the key role of pre-synaptic PLA_2_ neurotoxins in the pathophysiology of *D. siamensis* envenoming, Varespladib may be an alternative to, or an adjunct therapy to, the conventionally used antivenoms. Our previous work on Chinese *D. siamensis* venom demonstrated that Varespladib can partially reverse in vitro pre-synaptic neuromuscular inhibition when antivenom cannot [22]. Hence, it was of interest to study the efficacy of Varespladib against *D. siamensis* venoms from Thailand and Indonesia.

The role of antivenom in the treatment of envenomed patients is unequivocal. The absence of a specific antivenom against Indonesian *D. siamensis* venom underscores the need to explore other antivenoms or alternative treatments. Firstly, we investigated the potency of *D. siamensis* venom sourced from Thailand and Indonesia, specifically the Javan population. Secondly, we investigated the neutralising efficacy of two antivenoms from Thailand, in the presence and absence of Varespladib, to explore the potential complementarity of small molecule inhibitors with antivenoms, especially in cases where antivenom efficacy may prove suboptimal.

## 2. Results

### 2.1. Neurotoxic Effects of Thai and Javanese D. siamensis Venoms

Both Thai (3 and 10 µg/mL; TV; Figure 1a) and Javanese (3 and 10 µg/mL; JV; Figure 1c) *D. siamensis* venoms caused inhibition of indirect twitches of the chick biventer cervicis nerve–muscle preparation. Neither venom significantly inhibited responses to acetylcholine (ACh), carbachol (CCh), or potassium chloride (KCl) (Figure 1b,d), indicative of pre-synaptic neurotoxicity. The times to reach 90% inhibition (t_90_) of indirect twitches for the venom from Thailand were 58 ± 7 min and 41 ± 3 min for 3 µg/mL and 10 µg/mL, respectively. For the venom from Java, the t_90_ were 73 ± 4 min and 41 ± 2 min for 3 µg/mL and 10 µg/mL, respectively.

### 2.2. Myotoxic Effects of Thai and Javanese D. siamensis Venoms

Both Thai (10 and 30 µg/mL; TV; Figure 2a,b) and Javanese (10 and 30 µg/mL; JV; Figure 2c,d) *D. siamensis* venoms inhibited direct twitches, with a corresponding increase in baseline tension, over 180 min. This effect was not concentration-dependent.

### 2.3. Neurotoxicity: Antivenom Protection Studies

The prior addition of Thai *D. siamensis* monovalent antivenom (TMAV, at the recommended concentration) abolished the reduction in indirect twitches for both Thai (Figure 3a) and Javanese (Figure 3b) *D. siamensis* venoms. In contrast, the prior addition of non-specific Thai-neuro-polyvalent antivenom (TNPAV, 40 µL/mL) was markedly less effective, only delaying the inhibitory effects but not preventing them.

### 2.4. Neurotoxicity: Antivenom Reversal Studies

Thai *D. siamensis* monovalent antivenom (TMAV, 2× recommended concentration), added at either 30 or 60 min post venom, did not reverse indirect twitch inhibition caused by Thai *D. siamensis* venom (TV; 10 µg/mL) (Figure 4a).

Thai neuro-polyvalent antivenom (TNPAV, 40 µL/mL), added 30 min post venom, caused a small but significant delay in indirect twitch inhibition, although twitches were still abolished. This effect was not observed when TNPAV (40 µL/mL) was added 60 min after venom (Figure 4b).

Neither TMAV (2× recommended concentration; Figure 4c) nor TNPAV (40 µL/mL; Figure 4d) had a significant effect on indirect twitch inhibition when added 30 or 60 min after Javanese *D. siamensis* venom (JV; 10 µg/mL).

### 2.5. Washing Reversal Studies

Physical removal of venom from the organ bath via repeated washing with physiological salt solution at 30 or 60 min after the addition of venom did not reverse indirect twitch inhibition caused by either venom (Figure 5).

### 2.6. Varespladib Protection Studies

Pre-incubation (20 min) with Varespladib (0.8, 4 or 26 µM) prevented the pre-synaptic neurotoxicity induced by both (Figure 6a) Thai *D. siamensis* venom (TV; 10 µg/mL) and (Figure 6b) Javanese *D. siamensis* venom (JV; 10 µg/mL).

### 2.7. Varespladib Reversal Studies

The addition of Varespladib 60 min post venom caused a concentration-dependent restoration of indirect twitches, with ~50% reversal of inhibition for Thai *D. siamensis* venom (10 µg/mL; Figure 7a), and ~70% for Javanese *D. siamensis* venom (10 µg/mL; Figure 7c) at a Varespladib concentration of 26 μM.

In additional experiments, Varespladib (26 µM) was added three times (i.e., 60, 120, and 180 min) after the venom. However, the cumulative effect of three additions was not significantly different to the effect of a single concentration of Varespladib (26 µM) at 60 min for either venom (Figure 7b,d).

### 2.8. Antivenom and Varespladib Combination Prevention Studies

To determine the neutralising efficacy of the combination of antivenom and Varespladib, the concentrations of each treatment were lowered to induce partial neutralisation individually. Thai *D. siamensis* monovalent antivenom (0.25× the recommended concentration) and Varespladib (100 nM) caused partial protection against twitch inhibition induced by either Thai or Javanese *D. siamensis* venoms when added individually prior to venoms (Figure 8). The quantity of Thai neuro-polyvalent antivenom (40 µL/mL) was not lowered, as it was found to be only partially protective (see Figure 3).

For the combination protection protocol, antivenom was added to the preparation prior to the pre-incubation of venom with Varespladib (100 nM). The combination of Thai *D. siamensis* monovalent antivenom (0.25× the recommended concentration) or Thai neuro-polyvalent antivenom (40 µL/mL) with Varespladib (100 nM) markedly attenuated the pre-synaptic neurotoxicity induced by both Thai (TV; Figure 8a,b) and Javanese (JV; Figure 8c,d) *D. siamensis* venoms.

### 2.9. Antivenom and Varespladib Combination Reversal Studies

To examine the ability of antivenom (either Thai *D. siamensis* monovalent antivenom or Thai neuro-polyvalent antivenom) in combination with Varespladib (0.8 μM) to reverse the pre-synaptic neurotoxicity induced by Thai (TV; 10 μg/mL) or Javanese (JV; 10 μg/mL) *D. siamensis* venom, the combination treatment was added 60 min post venom. Given the antivenoms were shown to be ineffective at reversing the neurotoxicity, both antivenoms were used at the normal concentration (see Section 2.4).

The combination of Thai monovalent antivenom and Varespladib, added 60 min after venom, resulted in the reversal of indirect twitches that was statistically greater than the effect of Varespladib (0.8 µM) alone (Figure 9a,c). However, the combination of Thai neuro-polyvalent antivenom and Varespladib did not reverse the twitch inhibition more than Varespladib alone (Figure 9b,d).

## 3. Discussion

We examined the neuromuscular blocking effects of two Asian *D. siamensis* venoms from Thailand and Java (Indonesia) in the indirectly stimulated chick biventer cervicis nerve–muscle preparation. Both venoms were shown to exert potent pre-synaptic neurotoxicity, as evidenced by their t_90_ values (at 10 µg/mL) of 41 ± 3 min (Thai) and 41 ± 2 min (Javanese), respectively. No evidence of post-synaptic neurotoxicity was observed, as indicated by the absence of an inhibitory effect against the skeletal muscle nicotinic receptor agonists acetylcholine and carbachol. The pre-synaptic effects of both Thai and Javanese venoms were more potent than the venom from Chinese *D. siamensis* [21,22] and the venom of Sri Lankan *D. russelii* [8]. However, interestingly, clinically evident neurotoxic effects, i.e., ptosis, are more frequently seen in patients envenomed by the Sri Lankan species [25]. Both Thai and Javanese venoms also exhibited mild myotoxic effects, which were also more potent than the in vitro myotoxicity caused by Sri Lankan *D. russelii* venom, despite muscle damage being relatively absent following envenoming by Thai *D. siamensis* [18,39]. It is unclear whether Javanese *D. siamensis* envenoming involves myotoxic effects, but it was previously suggested that *D. siamensis* venoms from the Lesser Sunda (Indonesia) populations were reported to result in more severe local envenoming, including local tissue destruction and necrosis, which would suggest differences in the clinical symptomology of Russell’s viper populations across Indonesia [1,14].

Aside from Sri Lankan *D. russelii* and some Indian variants, neurotoxicity is not often reported following *D. siamensis* envenoming or is infrequent with some species [7,8,12]. Clinical neurotoxicity is either not well reported or not well described following envenoming by Thai or Javanese *D. siamensis*. It is well established that across the Asian *Daboia* genus, PLA_2_ neurotoxins are responsible for pre-synaptic neurotoxicity. Previously, Lingam et al. (2020) [16] reported a high abundance of PLA_2_ toxins in *D. siamensis* venoms from Thailand (37.92%) and Indonesia (48.37%), both of which resemble the venomic profile of *D. siamensis* from Taiwan, with the neurotoxic PLA_2_ components RV-4 and RV-7 [16,26,40].

Based on our data, both Thai and Javanese *D. siamensis* venoms were more potent than the viperid *Lachesis muta muta* (Southern American bushmaster) (t_50_: 30.4 ± 2.3 min) [41] and crotalids such as *Bothrops insularis* (Golden lancehead) (t_50_: 30 ± 1.9 min) [42] and *Bothrops neuwiedi* (Neuwied’s lancehead) (t_50_: 42 ± 2 min) [42] while being less potent than *Crotalus durissus terrificus* venom (South American rattlesnake), with a t_50_ of 16.3 ± 0.7 min [43]. These data indicate that both the Thai and Javanese *D. siamensis* venoms are some of the more potent pre-synaptic viperid venoms. In addition, the pre-synaptic neurotoxins PLA_2_ RV-7 and RV-4, a heterodimeric complex first identified in Taiwanese *D. siamensis* venoms [26,40], were also detected in both Thai and Javanese *D. siamensis* venoms following comparative proteomic studies, albeit the low levels of RV-4 in the two venoms are suspected to be the cause of a lack of evident neurotoxicity following *D. siamensis* envenoming [16]. There are examples where, despite the presence of pre-synaptic neurotoxins, venoms do not cause neurotoxicity in humans. The brown snake paradox refers to the lack of neurotoxicity following *Pseudonaja textilis* (Australian Eastern Brown snake) envenoming, despite the presence of the pre-synaptic neurotoxin textilotoxin, and seems to be due to the low abundance (i.e., 5.7%) of this relatively weak (t_50_: ~180 min) neurotoxin, together with differences in the actions of neurotoxins at the receptors in humans and animals [44]. Our findings suggest that the absence of neurotoxicity following *D. siamensis* envenoming is more likely attributable to a species-dependent effect rather than a quantitative difference given the high abundance [16,44] and potency of the toxins in the venoms examined in the current study. In fact, our prior investigations indicated that the pre-synaptic neurotoxin, U1-viperitoxin-Dr1a, from Sri Lankan *D. russelii* venom constitutes nearly 19% of this venom [8]. Despite its marked abundance, this neurotoxin exhibited low potency in an in vitro skeletal muscle preparation [8]. Paradoxically, the observed neurotoxicity following envenoming by Sri Lankan *D. russelii* is correlated with high venom concentrations [25]. The juxtaposition of quantity and intrinsic potency is a critical determinant of the onset of paralysis in humans. However, the generalisability of this paradigm to *D. siamensis* from Thailand and Indonesia remains ambiguous given the scarcity of well-defined neurotoxic symptoms in humans envenomed by these species. However, the neurotoxic actions of venoms/toxins in in vitro skeletal muscle preparations might not necessarily correlate with obvious neurotoxic envenoming in patients.

The pre-addition of Thai *D. siamensis* monovalent antivenom was able to protect against the pre-synaptic activity of both *D. siamensis* venoms, which was not unexpected since they have been shown to share similar venom profiles and likely share similar neurotoxins [16]. Our results align with prior findings demonstrating that Thai *D. siamensis* monovalent antivenom exhibits comparable immunoreactivity to both Thai and Indonesian venom fractions. Moreover, monovalent antivenom effectively reduces lethality in mice and mitigates the in vitro pro-coagulant effects caused by both Thai and Indonesian *D. siamensis* venoms on human citrated plasma [15,16]. Interestingly, Thai neuro-polyvalent antivenom, raised against elapid species venoms (i.e., *Ophiophagus hannah*, *Naja kaouthia, Bungarus candidus*, and *Bungarus fasciatus*), was able to partially protect against the pre-synaptic neurotoxic effects of *D. siamensis* venom. There has not been much investigation into the effects of Thai neuro-polyvalent antivenom against viperid venoms [45]. However, it was previously suggested that the cross-neutralising ability of Thai neuro-polyvalent antivenom against pre-synaptic toxins was a result of common antigenic regions shared across species [28,46].

The efficacy of the locally produced SABU (Serum Anti Bisa Ular), compared with Thai polyvalent antivenoms, was examined against various Indonesian snakes [27]. SABU was found to be less effective than Thai neuro-polyvalent antivenom in mice against the lethal effects of both homologous (included in the immunogen mixture) and heterologous (not included in the immunogen mixture) Indonesian snake venoms [27]. This suggests that the toxins in Thai and Indonesian *D. siamensis* venoms, particularly pre-synaptic neurotoxins, share epitopes with toxins from elapid species that can be neutralised by Thai neuro-polyvalent antivenom. Although in our study Thai neuro-polyvalent antivenom was less efficacious compared to the specific Thai monovalent antivenom, this supports our findings that the Thai antivenoms have good efficacy against Javanese *D. siamensis* venom.

Having confirmed the efficacy of the Thai antivenoms in prevention studies against the in vitro pre-synaptic neurotoxic effects of Thai and Javanese *D. siamensis* venoms, we then examined the ability of the antivenoms to reverse pre-synaptic neurotoxicity. Neither antivenom was able to reverse the pre-synaptic effects of either geographical variant, which confirms that pre-synaptic neurotoxicity is virtually irreversible with antivenom administration experimentally [36,47]. Thai neuro-polyvalent antivenom slightly delayed, but did not prevent, the effects of Thai *D. siamensis* venom when added at the earlier time point (i.e., 30 min post venom), suggesting that some epitopes may be shared between viperid (specifically Thai *D. siamensis*) and elapid venom toxins. However, this effect was not seen with Javanese *D. siamensis* venom and may be due to some geographical differences in venom components.

Removal of the venom from the organ bath, either 30 or 60 min after addition, followed by continued washing, failed to delay or prevent the subsequent abolishment of indirect twitches. These data are similar to those of an earlier work examining the reversal of the in vitro pre-synaptic neurotoxic effects of coastal taipan venom [47]. Our results further substantiate the hypothesis that PLA_2_ toxins are prone to internalisation or irreversible binding [37].

Our recent study showed that the pre-synaptic neurotoxicity caused by Chinese *D. siamensis* venom can be partially reversed by the post venom addition of Varespladib (0.8–26 µM) [22]. In the current study, the same concentrations of Varespladib pre-incubated with the venoms prevented the neurotoxicity induced by both Thai and Javanese *D. siamensis* venoms. This protective ability against PLA_2_-mediated toxicity, particularly pre-synaptic neurotoxicity, has been reported in skeletal muscle preparations and in vivo animal models against a variety of neurotoxic snake venoms [31,32,34,36,37,38,48,49,50]. Moreover, in the current study, the addition of Varespladib after both Thai and Javanese *D. siamensis* venoms reversed the neurotoxicity to a larger extent than seen with Chinese *D. siamensis* [22], *C. d. terrificus* [48], or *O. scutellatus* [37] venoms, with ~70% recovery for the Javanese *D. siamensis* venom. It has been proposed that Varespladib reduces the PLA_2_ toxicity by interfering with and binding to critical active sites required for function or blocking neurotoxins both inside and outside of the nerve terminals [37,51,52,53]. The marked reversal produced by Varespladib is interesting, as antivenom was ineffective when added at the same time point (i.e., 60 min after venom) and this suggests that aspects of pre-synaptic blockade are more reversible than previously thought.

The lack of full recovery of twitches using Varespladib could be due to irreversible damage to the motor nerve terminal and/or skeletal muscle. This residual inhibition does not appear to be due to insufficient Varespladib, as additional concentrations of Varespladib added after the initial concentration failed to produce further reversal. To substantiate these findings, it is important to extend our investigations to in vivo studies. These would facilitate a more comprehensive exploration into the capacity of Varespladib to aid recovery from pre-synaptic envenoming.

In circumstances where local or specific antivenoms are unavailable, non-specific antivenoms are often used. However, the antigenic activity of antivenoms may be reduced due to the taxonomic diversity of snake species and different venom compositions due to ontogenetic changes or geographical variations [52]. This can lead to poor clinical outcomes. In this context, adjunct inhibitors may improve the efficacy of even non-specific antivenoms. We initially explored the efficacy of combining monovalent or polyvalent antivenom with Varespladib in ‘prevention’ studies to validate the neutralising effects of these combinations. The combination of Thai *D. siamensis* monovalent antivenom or Thai neuro-polyvalent antivenom with Varespladib prevented the venom-induced neurotoxicity caused by either Thai or Javanese *D. siamensis* venom. Notably, the combination of Varespladib and Thai neuro-polyvalent antivenom was markedly more effective than the non-specific antivenom alone in preventing twitch inhibition. As the combination treatments were able to neutralise the neurotoxic activity of the *D. siamensis* venoms, we examined whether the combination could reverse the inhibition of twitch activity better than Varespladib alone given that the antivenoms were completely ineffective alone when administered after venom.

The combination of Thai *D. siamensis* monovalent antivenom with a low concentration of Varespladib resulted in additional recovery of twitches, better than that of either treatment alone for both venoms. This contrasts with the combination studies we previously performed with Chinese *D. siamensis* venom, where the combination of specific antivenom and Varespladib did not result in enhanced reversal [22]. This phenomenon is intriguing since antivenoms are generally acknowledged to be ineffective against pre-synaptic neurotoxicity. In contrast, the combination of Varespladib and Thai neuro-polyvalent antivenom did not produce enhanced recovery compared with Varespladib alone. It is proposed that due to their small molecular weight and charge, small molecule therapeutics (e.g., Varespladib) typically exhibit larger volumes of distribution (Vd) and enhanced tissue penetration compared to IgG antibodies [54,55]. Consequently, incorporating penetrating molecules could potentially prolong the effectiveness of antivenoms. Considering the distinctive characteristics of small molecule inhibitors and antibodies, future investigations may focus on exploring the interactions between antivenom and Varespladib. This may involve assessing the pharmacodynamics and pharmacokinetics characteristics in vivo, as well as confirming the dose selection or regimen optimisation.

## 4. Conclusions

In conclusion, our observations highlight the shared pre-synaptic neurotoxicity of *D. siamensis* venom sourced from Thailand and Java. Notably, the neurotoxic activity persists, despite the supposedly marked divergence of the Javan population from its remaining counterparts in the Asiatic mainland. Furthermore, we have shown the cross-reactivity of Thai antivenoms against the venom of Javanese *D. siamensis*. This reveals a plausible efficacy in addressing snakebite incidents in Indonesia, especially in instances where skepticism surrounds the effectiveness of the locally produced antivenom. In addition, Varespladib has potential to help relieve the need for antivenom or improve the overall efficacy of antivenoms, particularly where delays in treatment are unavoidable.

## 5. Materials and Methods

### 5.1. Animals

Five- to ten-day-old male brown chicks (White Leghorn crossed with New Hampshire) were obtained from Wagner’s Poultry, Coldstream, Victoria (Australia). The animals were given free access to food and water.

### 5.2. Chemicals and Drugs

The following drugs were purchased from Sigma-Aldrich, St Louis, MO, USA: acetylcholine chloride, carbamylcholine, potassium chloride, d-tubocurarine, bovine serum albumin (BSA), Varespladib (CAS: 172732-68-2), and dimethyl sulfoxide (DMSO). All the drugs were dissolved in Milli-Q water, with the exception of Varespladib, which was dissolved in DMSO to a final concentration of 10 mM and then subsequently diluted in physiological solution.

### 5.3. Venoms and Antivenoms

The venoms of the *D. siamensis* species were obtained from the following locales: (1) Bangkok, Thailand, from captive species (>20 specimens of both male and female) held at QSMI, Thai Red Cross Society; (2) Java, Indonesia, gifted from Venom Supplies, Australia. All the venoms were pooled from several adult snakes of both sexes, freeze-dried, and stored at −20 °C prior to use. The venoms were dissolved in 0.05% BSA and stored at 4 °C for use. Thai *D. siamensis* monovalent antivenom (LOT WR00121, MFG: 19/2/2021, Exp: 19/2/2026) and Thai neuro-polyvalent antivenom (LOT NP00120, MFG: 27/3/2020, Exp: 27/3/2025) were purchased from QSMI, Thai Red Cross Society, Bangkok, Thailand. The freeze-dried antivenoms were reconstituted with the supplied solution from the manufacturer. The amount of each antivenom required to neutralise venom was based on the neutralisation ratio stated by the manufacturer. According to the instructions, 1 mL of Thai *D. siamensis* monovalent antivenom neutralises 0.6 mg *D. siamensis* venom. When this proportion (amount of antivenom calculated to neutralize 10 µg/mL; 1×) was tested in preliminary protection studies, complete neutralisation was observed. For this reason, we used no less than the standard volume of 1× the recommended antivenom volume. For neuro-polyvalent antivenom, 1 mL of Thai antivenom neutralises the following: 0.8 mg *Ophiophagus hannah*, 0.6 mg *Naja kaouthia*, 0.4 mg *Bungarus candidus*, and 0.6 mg *Bungarus fasciatus* venom. To achieve sufficiently high concentrations of antivenom against venoms which are not included in the immunogen mixture, 40 µL/mL of TNPAV was used.

### 5.4. Chick Biventer Cervicis Nerve–Muscle Preparation

Male brown chicks were humanely euthanised following CO_2_ inhalation. From each chick, two nerve muscles were dissected and mounted vertically onto wire holders under a 1 g resting tension in a 5 mL organ bath. The preparations were maintained in 34 °C physiological solution (composition: 118.4 mM of NaCl, 25 mM of NaHCO_3_, 11.1 mM of glucose, 4.7 mM of KCl, 1.2 mM of MgSO_4_, 1.2 mM of KH_2_PO_4_ and 2.5 mM of CaCl_2_) and constantly aerated with carbogen (95% O_2_ and 5% CO_2_). The motor nerves of the preparations were indirectly stimulated with a supramaximal voltage of 10–15 V at a frequency of 0.1 Hz and a duration of 0.2 ms using an electrical stimulator. The corresponding twitches were recorded using a PowerLab system (ADInstruments Pty Ltd. Bella Vista, NSW, Australia) via a Grass FT03 force transducer. The tissues were allowed to equilibrate and stabilise for 20 min before the addition of 10µM dTC (Sigma-Aldrich). The abolishment of twitches using dTC confirmed selective stimulation of the nerves rather than the muscle. The preparations were then repeatedly washed with physiological solution to restore twitches. In the absence of electrical stimulation, contractile responses to exogenous ACh (1 mM, 30 s), CCh (20 µM, 60 s), and KCl (40 mM, 30 s) were obtained, with washing and returns to baseline between each addition. Following this, electrical stimulation was recommenced for at least 20–30 min until steady twitch heights were obtained. For directly evoked twitches, the muscle belly was stimulated with a supramaximal voltage of 15–20 V at a frequency of 0.1 Hz and a duration of 2 ms. Nerve stimulation was abolished by the addition of 10 µM dTC which remained in the bath for the duration of the experiment.

The neutralisation effects of antivenom and/or Varespladib (CAS: 172732-68-2) were evaluated using either a protection or reversal protocol. (1) Pre-incubation of venoms (3, 10 or 30 µg/mL) in the absence of treatment; (2) addition of antivenom (added at the corresponding neutralisation ratio indicated by the manufacturer) 20 min prior to the addition of venom or pre-incubation with Varespladib for 20 min (100 nM–26 µM) before addition of the mixture to the organ bath; (3) antivenom was added to the bath prior to the addition of venom already pre-incubated with Varespladib (20 min); (4) reversal protocol where washing was commenced, or antivenom and/or Varespladib was added 30 or 60 min after venom. The concentrations of Varespladib used in this study were adapted from the protocol described in a previous study and represent the range of previously validated concentrations [22]. Notably, our lowest concentration of Varespladib (100 nM) is lower than the previously tested concentrations [22]. Twitch activity was observed for a further 3 h after treatment addition. At the conclusion of all experiments, electrical stimulation was ceased, and ACh, CCh, and KCl were re-added (as above).

### 5.5. Data Analysis and Statistics

Twitch height was measured from baseline every 4 min after venom addition and expressed as a percentage of the pre-venom twitch height. The post venom contractile responses to ACh, CCh, and KCl were expressed as a percentage of the corresponding pre-venom contractile response. In the myotoxicity study, changes in baseline tension were measured every 10 min after venom addition. One-way analysis of variance (ANOVA) was performed for comparisons between different treatments or the control. Comparisons of responses to agonists pre- and post venom were performed using Student’s paired t-test. All ANOVAs were followed by Bonferroni multiple comparison. All data are represented as the mean ± standard error of the mean (SEM) where n is the number of tissue preparations. All the analyses were performed using GraphPad Prism 10.1.2, and *p* < 0.05 was considered statistically significantly.

## Figures and Tables

**Figure 1 toxins-16-00124-f001:**
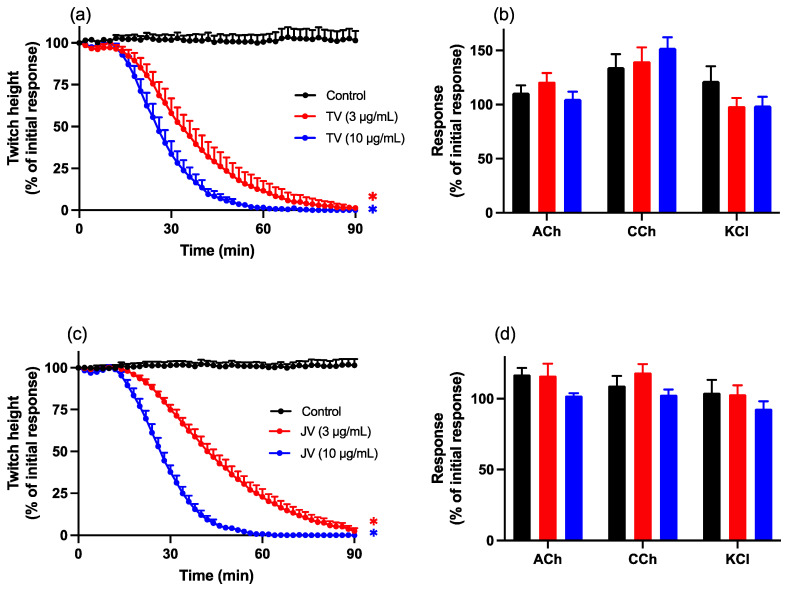
Effect of Thai (TV) and Javanese (JV) *D. siamensis* venoms on (**a**,**c**) indirect twitches or (**b**,**d**) contractile responses to the agonists acetylcholine (ACh), carbachol (CCh), and potassium chloride (KCl) in the chick biventer cervicis nerve–muscle preparation. Data presented as the mean ± SEM; * *p* < 0.05, significantly different from vehicle control (Control) at 90 min; one-way ANOVA followed by Bonferroni multiple comparison post hoc test, *n* = 6.

**Figure 2 toxins-16-00124-f002:**
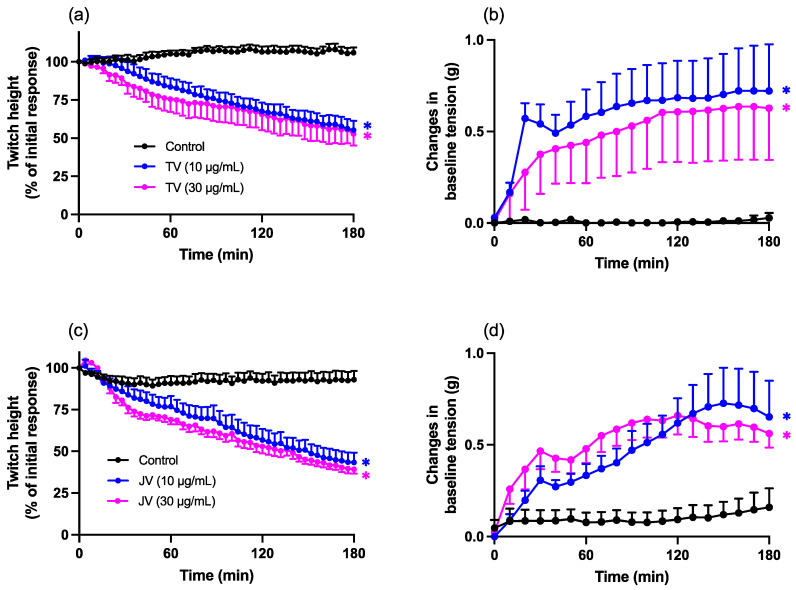
Effect of Thai (TV) and Javanese (JV) *D. siamensis* venoms on (**a**,**c**) direct twitches or (**b**,**d**) baseline tension in the chick biventer cervicis nerve–muscle preparation. Data presented as the mean ± SEM; * *p* < 0.05, significantly different from vehicle control (Control) at 180 min; one-way ANOVA followed by Bonferroni multiple comparison post hoc test, *n* = 6.

**Figure 3 toxins-16-00124-f003:**
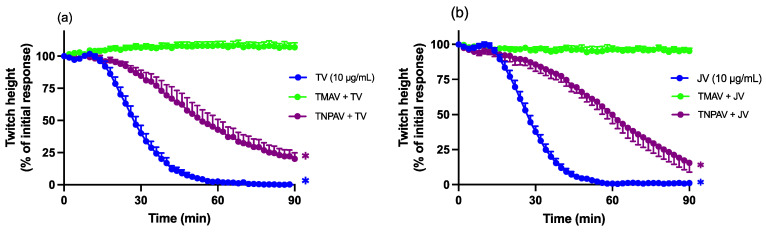
Effect of prior addition (20 min) of Thai *D. siamensis* monovalent antivenom (TMAV) or Thai neuro-polyvalent antivenom (TNPAV) on pre-synaptic neurotoxicity induced by (**a**) Thai *D. siamensis* venom (TV; 10 µg/mL) and (**b**) Javanese *D. siamensis* venom (JV; 10 µg/mL). Data presented as the mean ± SEM; * *p* < 0.05, significantly different from respective antivenom control (i.e., antivenom alone) at 90 min; one-way ANOVA followed by Bonferroni multiple comparison post hoc test, *n* = 4–6.

**Figure 4 toxins-16-00124-f004:**
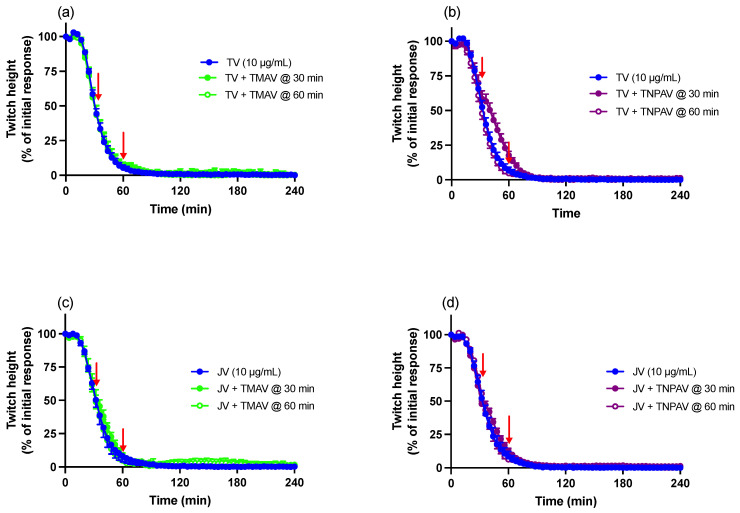
The effect of Thai *D. siamensis* monovalent antivenom (TMAV; 2× the recommended concentration) or Thai neuro-polyvalent antivenom (TNPAV; 40 µL/mL), added 30 or 60 min post venom, on the pre-synaptic neurotoxic effects of (**a**,**b**) Thai *D. siamensis* venom (TV; 10 µg/mL) and (**c**,**d**) Javanese *D. siamensis* venom (JV; 10 µg/mL). Red arrows indicate the addition of antivenom. Data presented as the mean ± SEM, *n* = 4–6.

**Figure 5 toxins-16-00124-f005:**
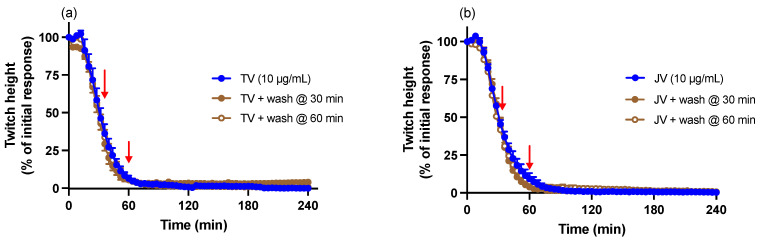
Effect of repeated washing, commencing 30 or 60 min post venom, on (**a**) Thai *D. siamensis* venom (TV; 10 µg/mL)- or (**b**) Javanese *D. siamensis* venom (JV; 10 µg/mL)-induced pre-synaptic neurotoxicity. Red arrows indicate the commencement of washing. Data presented as the mean ± SEM, *n* = 4–6.

**Figure 6 toxins-16-00124-f006:**
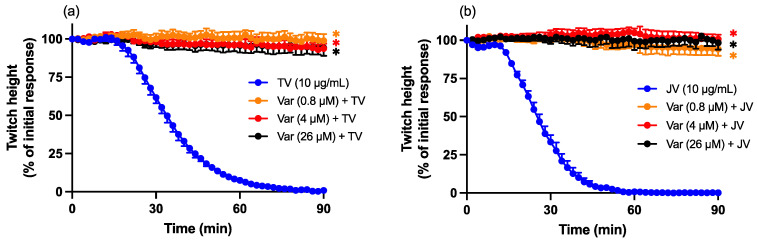
Protective effects of Varespladib (0.8, 4 or 26 µM) against (**a**) Thai *D. siamensis* venom (TV; 10 µg/mL)- or (**b**) Javanese *D. siamensis* venom (JV; 10 µg/mL)-mediated neurotoxicity. Data presented as the mean ± SEM; * *p* < 0.05, significantly different from venom alone at 90 min; one-way ANOVA followed by Bonferroni multiple comparison post hoc test, *n* = 4–6.

**Figure 7 toxins-16-00124-f007:**
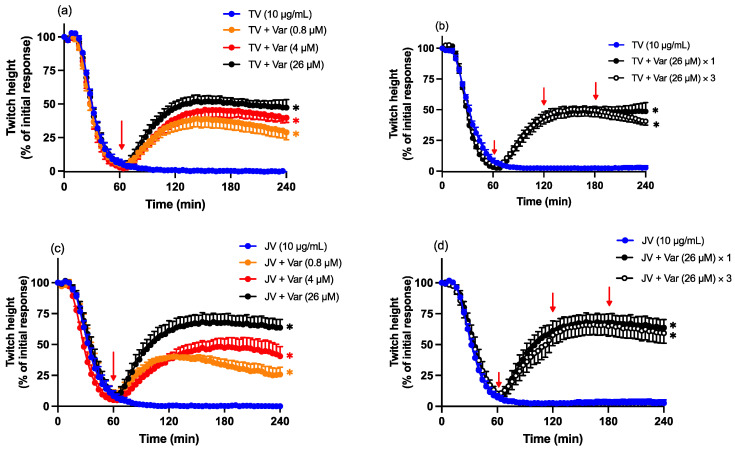
Effect of single (60 min) or multiple (60, 120 and 180 min) additions of Varespladib (Var) after venom on (**a**,**b**) Thai (TV; 10 µg/mL) or (**c**,**d**) Javanese (JV; 10 µg/mL) *D. siamensis* venom-mediated pre-synaptic neurotoxicity. Red arrows indicate the addition of Varespladib. Data presented as the mean ± SEM; * *p* < 0.05, significantly different to venom alone at 240 min; one-way ANOVA followed by Bonferroni multiple comparison post hoc test, *n* = 4–6. NB. The Varespladib (26 μM × 1) data in (**a**) are reproduced in (**b**), and the Varespladib (26 μM × 1) data in (**c**) are reproduced in (**d**) for ease of comparison.

**Figure 8 toxins-16-00124-f008:**
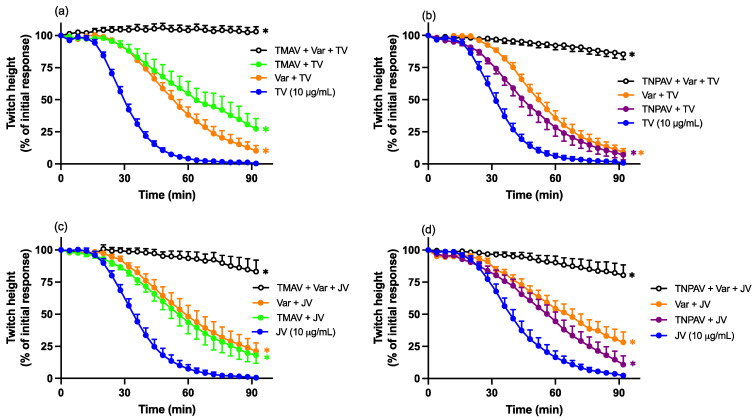
Effect of the pre-addition of the combination of Thai monovalent antivenom (TMAV; 0.25× the recommended concentration) or Thai neuro-polyvalent antivenom (TNPAV; 40 µL/mL) and Varespladib (Var; 100 nM) on (**a**,**b**) Thai (TV; 10 µg/mL)- or (**c**,**d**) Javanese (JV; 10 µg/mL)-venom-mediated pre-synaptic neurotoxicity. Data presented as the mean ± SEM; * *p* < 0.05, significantly different to venom alone at 90 min; one-way ANOVA followed by Bonferroni multiple comparison post hoc test, *n* = 6.

**Figure 9 toxins-16-00124-f009:**
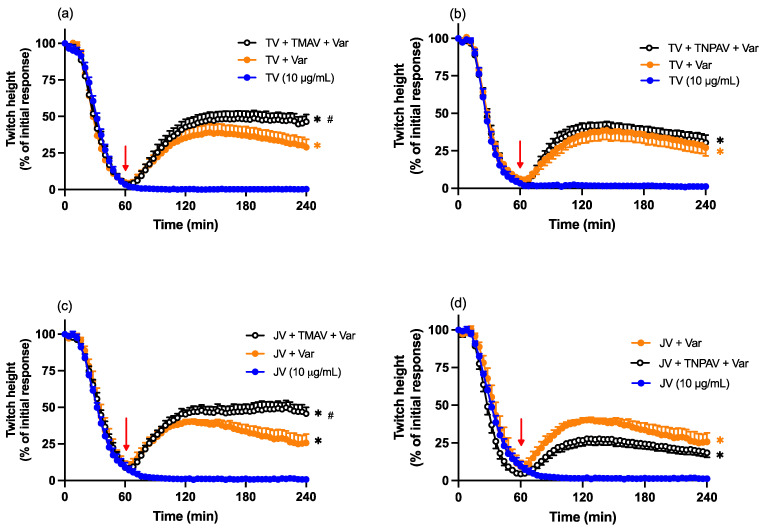
Effect of the combination of Thai monovalent antivenom (TMAV; 2× the recommended concentration) or Thai neuro-polyvalent antivenom (TNPAV; 40 µL/mL) with Varespladib (Var; 0.8 µM), added 60 min post venom, on (**a**,**b**) Thai (TV; 10 µg/mL)- or (**c**,**d**) Javanese (JV; 10 µg/mL)-venom-mediated pre-synaptic neurotoxicity. Red arrows indicate the addition of combination treatment. Data presented as the mean ± SEM; * *p* < 0.05, significantly different to venom alone at 240 min; # *p* < 0.05, significantly different to Varespladib (0.8 µM) alone at 240 min; one-way ANOVA followed by Bonferroni multiple comparison post hoc test, *n* = 6.

## Data Availability

Data are contained within the article.

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
