# Peer review of "A Comparison of the Efficacy of Antivenoms and Varespladib against the In Vitro Pre-Synaptic Neurotoxicity of Thai and Javanese Russell’s Viper (Daboia spp.) Venoms"

_toxins, 2024, doi:10.3390/toxins16030124_

Round 1

Reviewer 1 Report

Comments and Suggestions for Authors

General Comments:

I am grateful for the opportunity to review the comprehensive study entitled "A Comparison of the Efficacy of Antivenoms and Varespladib against In Vitro Pre-synaptic Neurotoxicity of Thai and Javanese Russell’s Viper (Daboia spp.) Venoms." The authors have presented a meticulously organized and clearly articulated paper review. Their review is thorough, encapsulating the relevant literature on the subject matter comprehensively. The primary objective of this study was to elucidate the neuromuscular activity of Daboia siamensis venom from Thailand and Java (Indonesia) and to assess the efficacy of Thai antivenoms and/or Varespladib in preventing or reversing these effects.

However, snakebite envenoming remains a significant health concern particularly in Indonesia, leading to considerable morbidity and mortality with an estimated 10,000 to 15,000 cases annually. Although the incidence appears to be on the rise, accurately quantifying fatalities remains challenging. 

The critical need for specific antivenoms is underscored in Indonesia due to the prevalence of numerous medically significant snake species native to the region. The study categorizes Elapid venom (Daboia siamensis) as primarily neurotoxic, contrasting with viperid venoms, which are identified mainly for their cytotoxic and/or hemotoxic characteristics (defined here as toxicity targeting blood and the cardiovascular system, including hemostasis).

Remarkably, the Daboia siamensis monovalent antivenom (DSMAV, Thailand) demonstrated substantial immunoreactivity towards the venoms of eastern Russell's vipers from both Thailand and Indonesia. It effectively neutralized the procoagulant and lethal effects of both venoms, showcasing its high potency. Given these findings,

I believe this study can be published with minor corrections.

Author Response

No changes required. We thank the reviewer for their kind comments.

Reviewer 2 Report

Comments and Suggestions for Authors

Antivenoms remain the only treatment for managing the complications caused by lethal venoms, since its development more than a century ago. They constitute life-saving treatment regimens, but they also have some limitations. In this nice study, the authors assessed the ability of antivenoms to prevent and reverse the in vitro neurotoxicity of venoms from medically important snakes. In addition, the authors evaluated the potential of a promising next-generation antivenom varespladib, which interferes with the PLA2 activity of venom enzymes. This drug has a safety profile, and has been repurposed to mitigate the toxic effects of snake venom PLA2s. The findings are relevant for the design of future clinical trials and the development of novel antivenoms. In fact, there is an ongoing clinical trial (BRAVO) assessing the benefits of varespladib in India and the USA.  The results are clear and well-presented. I do recommend the publication of this interesting study, but before I encourage the authors to address some major and minor points. The major points are related to some clarifications in the methodology section and mainly rely on the molar equivalence of antivenoms and varespladib. In a comparison study, it is important to ensure that the number of molecules is similar, mainly when they are significantly different in molecular weight. If the evaluation did not consider it, the authors should include the limitations.

Major points

1. How was the amount of antivenom used determined? Is this concentration closer to the dose used in hospital settings?  

2. Authors need to clarify if the amount of antivenom and varespladib used are equivalent in molar terms. This is really important, because they are establishing a comparative study. Immunoglobulin are big macromolecules, in comparison to the low molecular weight of small-molecule inhibitors, such as varespladib. So, it is possible that a greater number of molecules of inhibitors were used than immunoglobulins (please consider having a molar view).

3. How authors determine the concentration of varespladib needs to be justified. Is all PLA enzyme activity inhibited in this concentration? Did the authors perform some enzyme assays before to estimate the amount of varespladib?

4. The abstract is too long. The authors must summarise and emphasise the main findings. Future directions must be included. According to the journal's guidelines, the abstract section should be a total of about 200 words maximum. The current version contains more than the list of required words.

5. Why do the authors use e chick biventer cervicis nerve-muscle preparation? Would not be more relevant to use a mouse model? Some studies have shown that avian preparations are usually more susceptible to venom toxins than mammalian models. In this context, I do believe this manuscript (https://doi.org/10.1016/j.cbpc.2018.03.008) may be useful to enrich the discussion around this topic and to justify the preparation choice. 

Minor

1. The two last sentences of the abstract section are contradictory. How there is an enhancement of the ability of monovalent antivenom to reverse pre-synaptic neurotoxicity? The previous sentence clearly that monovalent antivenom does not have this property (no reversal was observed).

2. Key contribution. The authors should not extrapolate their findings. They focused on only one important effect of venoms. They can not translate this to snakebites, where different clinical manifestations are observed.

3. Some sentences need to be supported by references. For example, lines 43-44, 47-49, 51-53, 105-107 and so on.

4. Line49-51. The spectrum of clinical manifestations of snakebites can be wider than the traditional view or didactic explanation of viper and elapid envenomation. I am encouraging the authors to recognise it. This can benefit and help doctors and medical professionals working in the first line of diagnosis and treatment to identify some rare or underreported symptoms or pathological effects. This is particularly relevant in the context of the Daboia genus. The number of case reports detailing these less common events is increasing, alerting the need to create awareness around this topic. Here, there are two examples: https://doi.org/10.22038/apjmt.2021.18818, doi: 10.1016/j.toxicon.2023.107068. However, authors can also find other case reports.

5. Line 51. The variability in the clinical profile is more complex. It is not only because of the biochemical diversity of venoms but also because of the amount of venom, and bite site, among others. Venom composition is key, but it represents only one dimension.

6. The introduction section is quite long.  Authors can summarise, highlighting crucial points.

7. Figures 1 and 2. I do not understand how the statistical comparison was performed. Did the authors compare only the last time point? Why are the asterisks out of the curves? Were the data compared only with the control? Please clarify this point.

8. Lines 170-172, 188-191, and so on. Please avoid a one-sentence paragraph. Expand the idea or combine sentences with similar ideas.  

9. Please revise the possible high number of self-citations. I do not know who the authors are, because the journal operates a double-blind peer-reviewed process. However, I identified a high number of articles from Hodgson`s group being cited in the reference sections (approximately 20%). Please revise COPE guidelines and journal’s guidelines.

Author Response

Major points

  1. How was the amount of antivenom used determined? Is this concentration closer to the dose used in hospital settings?  

The amount of antivenom that we used is based on the neutralisation rate provided by the manufacturer. Every vial of antivenom states how many mL of antivenom will neutralise x amount of venom (mg). In this study, we used an amount of antivenom that would be able to neutralise the amount of venom used in the organ bath. This has been stated in the Methods section 5.3. This is referred to as “1x” recommended dose in the manuscript. We gradually increase the amount of AV if no protective effective is seen. The same strategy is applied to the clinical setting but estimating the amount of venom ‘injected’ into the patient and amount of venom in the circulation is more challenging.

  1. Authors need to clarify if the amount of antivenom and varespladib used are equivalent in molar terms. This is really important, because they are establishing a comparative study. Immunoglobulin are big macromolecules, in comparison to the low molecular weight of small-molecule inhibitors, such as varespladib. So, it is possible that a greater number of molecules of inhibitors were used than immunoglobulins (please consider having a molar view).

While we acknowledge this comment, we are not concerned about using the same M quantity of antivenoms and Varespladib. We are interested in using the amount of antivenom which has been calculated to neutralise the quantity of venom being used (as discussed above), as well as higher concentrations to see if there is a concentration-dependent effect, and an efficacious concentration of Varespladib. That is why we first study the preventative effective of the antivenoms and/or Varespladib. That is, can they neutralise the venom under ‘perfect’ conditions? If they cannot prevent neurotoxicity, they will never be able to reverse neurotoxicity. Success in this experiment confirms the ‘efficacy’ of the therapies. Whether they can reverse the effects of the venoms shows whether they are ‘effective’ under our experimental conditions and provides insight into whether they will be clinically effective.

  1. How authors determine the concentration of varespladib needs to be justified. Is all PLA enzyme activity inhibited in this concentration? Did the authors perform some enzyme assays before to estimate the amount of varespladib?

Our previous publication (doi:10.3390/toxins15010062) established the concentrations of Varespladib that were effective, or not, against in vitro neurotoxicity induced by Chinese Russell’s viper venom, which we have referenced in the current manuscript. However, given there are potency differences between this venom and the venoms being used in the current study, we have also tested other concentrations of Varespladib to determine which doses remain effective. There is clearly a marked difference between the effects of venom in the absence and presence of Varespladib, indicating that the PLA2 enzymes responsible for pre-synaptic neurotoxicity are inhibited. We have completed a PLA2 assay to determine whether the PLA2 activity is inhibited, but this was not included as it was a preliminary study to determine what starting concentration to use, and then the proceeding concentrations were then tested in the chick biventer cervicis nerve muscle preparation.

  1. The abstract is too long. The authors must summarise and emphasise the main findings. Future directions must be included. According to the journal's guidelines, the abstract section should be a total of about 200 words maximum. The current version contains more than the list of required words.

We have reduced the length of the abstract.

  1. Why do the authors use the chick biventer cervicis nerve-muscle preparation? Would not be more relevant to use a mouse model? Some studies have shown that avian preparations are usually more susceptible to venom toxins than mammalian models. In this context, I do believe this manuscript (https://doi.org/10.1016/j.cbpc.2018.03.008) may be useful to enrich the discussion around this topic and to justify the preparation choice. 

The chick biventer cervicis nerve muscle is preferred over the mouse preparation as it is the only skeletal muscle preparation that can distinguish between pre- and post-synaptic neurotoxicity. This is due to the presence of both focally- and multiply-innervated muscle fibres. This means the preparation can be electrically stimulated to induce twitches (i.e. focally-innervated) but also responds by slower contractions to external N receptor agonists (i.e. multiply-innervated fibres). This difference has been well described by us (and others) in a number of review articles.

Minor

  1. The two last sentences of the abstract section are contradictory. How there is an enhancement of the ability of monovalent antivenom to reverse pre-synaptic neurotoxicity? The previous sentence clearly that monovalent antivenom does not have this property (no reversal was observed).

Thank you for this comment. There is no reversal observed with monovalent antivenom alone, however in combination with Varespladib, there was enhancement in the ability to reverse pre-synaptic neurotoxicity. This has been clarified in the abstract.

  1. Key contribution. The authors should not extrapolate their findings. They focused on only one important effect of venoms. They cannot translate this to snakebites, where different clinical manifestations are observed.

We have amended this in the manuscript.

  1. Some sentences need to be supported by references. For example, lines 43-44, 47-49, 51-53, 105-107 and so on.

Lines 43-44 is a ‘key contribution’ and does not require a reference. Lines 47-49 is supported by references #1 and #2. Lines 51-53 is supported by references #3 to 7. Lines 105-107 is supported by reference #26 and it is of general knowledge (and our conclusion based on our background readings) that alternative therapies are needed and does not need a reference.

  1. Line49-51. The spectrum of clinical manifestations of snakebites can be wider than the traditional view or didactic explanation of viper and elapid envenomation. I am encouraging the authors to recognise it. This can benefit and help doctors and medical professionals working in the first line of diagnosis and treatment to identify some rare or underreported symptoms or pathological effects. This is particularly relevant in the context of the Daboia genus. The number of case reports detailing these less common events is increasing, alerting the need to create awareness around this topic. Here, there are two examples: https://doi.org/10.22038/apjmt.2021.18818, doi: 10.1016/j.toxicon.2023.107068. However, authors can also find other case reports.

Thank you for providing these. Our study focuses on the Eastern Daboia siamensis species, which are a separate species to Daboia russelii stated in the papers above. D. russelii has been recognised as the Western species and are known to exert different pathological effects to those of Daboia siamensis. It is also fair to mention that Dabaoia siamensis envenomation can also lead to unique pathological effects not seen in D. russelii. We have now included a brief section addressing this in the manuscript.  

  1. Line 51. The variability in the clinical profile is more complex. It is not only because of the biochemical diversity of venoms but also because of the amount of venom, and bite site, among others. Venom composition is key, but it represents only one dimension.

Thank for providing this insight. Yes, it is true that many other important factors contribute to the vast clinical profile. Our main point is that even within the same species, there is venom variation which can lead to differences in clinical profiles. More so, the effectiveness of antivenom is affected by the composition of venom. Even small differences in venom composition may have marked impact on antivenom effectiveness. Since we were looking at the differences in venom potency and antivenom effectiveness due to the geographical separation of D. siamensis in Thailand and Java-Indonesia, it was less relevant to include amount of venom and where the bite had originated. We have amended this in the manuscript to further clarify.

  1. The introduction section is quite long.  Authors can summarise, highlighting crucial points.

We believe that the current Introduction is of an appropriate length to cover the necessary background to the study. It would be difficult to shorten the Introduction without removing important information for the reader. In particular, we believe that it is vital to cover why we are studying D. siamensis venoms, the absence of specific antivenoms, the difficulties that emerge from this deficit in a clinical situation, and the potential for complementary therapies.  

  1. Figures 1 and 2. I do not understand how the statistical comparison was performed. Did the authors compare only the last time point? Why are the asterisks out of the curves? Were the data compared only with the control? Please clarify this point.

The statistical comparison was made between the last time points of each c-r response curve. We have clarified this in each figure legend.

  1. Lines 170-172, 188-191, and so on. Please avoid a one-sentence paragraph. Expand the idea or combine sentences with similar ideas.  

Amended as requested.

  1. Please revise the possible high number of self-citations. I do not know who the authors are, because the journal operates a double-blind peer-reviewed process. However, I identified a high number of articles from Hodgson`s group being cited in the reference sections (approximately 20%). Please revise COPE guidelines and journal’s guidelines.

We have added additional references (as requested in previous feedback), which has reduced the ‘self-citation’ percentage. However, as one of the leading groups in the world studying the neurotoxicity of snake venoms, it is virtually impossible not to quote a number of our previous papers. This % quantity of self-citations, used by Toxins, is a blunt tool and needs to be reviewed in the context of each manuscript. We believe that the manuscripts quoted are essential to the understanding of the data and putting our results into the broader context of the field.

Reviewer 3 Report

Comments and Suggestions for Authors

This is an excellent publication and continues to expand the body of knowledge of PLA2 inhibitors.

Author Response

(The authors gave the same response as above.)

Round 2

Reviewer 2 Report

Comments and Suggestions for Authors

The authors have carefully addressed the most relevant suggestions raised by the reviewers. The manuscript brings novelties to the snakebite and venom pharmacology fields. I recommend its publication, however there is an important point that I suggest the authors address in the discussion section.

1. I have now understood the rationale behind the choice of antivenom and enzyme inhibitors, however, this is also a limitation of the current study, which did not establish an equivalent molar comparison. Although the results are still important, inhibitors and immunoglobulins have very different molecular weights. Therefore, it is not surprising that small molecule inhibitors usually present better performance than antivenoms when compared in this scenario. Of course, this does not minimise the interesting findings of the manuscript, but readers and authors need to be careful in interpreting the results under this context. Future studies must consider the molar equivalence, and ensure the same number of molecules when the main purpose is the comparison of antivenom strategies. 

Author Response

Thanks you for this comment. We have added an additional paragraph at the end of the Discussion and an additional reference (#55) to the Reference section.